# The Oncolytic Adenovirus XVir-N-31, in Combination with the Blockade of the PD-1/PD-L1 Axis, Conveys Abscopal Effects in a Humanized Glioblastoma Mouse Model

**DOI:** 10.3390/ijms23179965

**Published:** 2022-09-01

**Authors:** Moritz Klawitter, Ali El-Ayoubi, Jasmin Buch, Jakob Rüttinger, Maximilian Ehrenfeld, Eva Lichtenegger, Marcel A. Krüger, Klaus Mantwill, Florestan J. Koll, Markus C. Kowarik, Per Sonne Holm, Ulrike Naumann

**Affiliations:** 1Molecular Neurooncology, Department of Vascular Neurology, Hertie Institute for Clinical Brain Research and Center Neurology, University of Tübingen, D-72076 Tübingen, Germany; 2Department of Urology, Klinikum Rechts der Isar, Technical University of Munich, D-81675 Munich, Germany; 3Department of Preclinical Imaging and Radiopharmacy, University of Tübingen, D-72076 Tübingen, Germany; 4B Cell Immunology, Department of Vascular Neurology, Hertie Institute for Clinical Brain Research and Center Neurology, University of Tübingen, D-72076 Tübingen, Germany; 5Department of Oral and Maxillofacial Surgery, Medical University Innsbruck, A-6020 Innsbruck, Austria; 6XVir Therapeutics GmbH, D-80331 Munich, Germany

**Keywords:** glioblastoma, oncolytic virotherapy, immune checkpoint inhibition, abscopal effects, XVir-N-31

## Abstract

Glioblastoma (GBM) is an obligatory lethal brain tumor with a median survival, even with the best standard of care therapy, of less than 20 months. In light of this fact, the evaluation of new GBM treatment approaches such as oncolytic virotherapy (OVT) is urgently needed. Based on our preliminary preclinical data, the YB-1 dependent oncolytic adenovirus (OAV) XVir-N-31 represents a promising therapeutic agent to treat, in particular, therapy resistant GBM. Preclinical studies have shown that XVir-N-31 prolonged the survival of GBM bearing mice. Now using an immunohumanized mouse model, we examined the immunostimulatory effects of XVir-N-31 in comparison to the wildtype adenovirus (Ad-WT). Additionally, we combined OVT with the inhibition of immune checkpoint proteins by using XVir-N-31 in combination with nivolumab, or by using a derivate of XVir-N-31 that expresses a PD-L1 neutralizing antibody. Although in vitro cell killing was higher for Ad-WT, XVir-N-31 induced a much stronger immunogenic cell death that was further elevated by blocking PD-1 or PD-L1. In vivo, an intratumoral injection of XVir-N-31 increased tumor infiltrating lymphocytes (TILs) and NK cells significantly more than Ad-WT not only in the virus-injected tumors, but also in the untreated tumors growing in the contralateral hemisphere. This suggests that for an effective treatment of GBM, immune activating properties by OAVs seem to be of greater importance than their oncolytic capacity. Furthermore, the addition of immune checkpoint inhibition (ICI) to OVT further induced lymphocyte infiltration. Consequently, a significant reduction in contralateral non-virus-injected tumors was only visible if OVT was combined with ICI. This strongly indicates that for an effective eradication of GBM cells that cannot be directly targeted by an intratumoral OV injection, additional ICI therapy is required.

## 1. Introduction

Glioblastoma (GBM) is the most common malignant primary brain tumor in adults with a devastating prognosis. Even with the latest multimodal therapy options, the median survival is less than 20 months [1]. The devastating prognosis and therapy resistance of GBM are provoked by the malignant characteristics as GBM cells infiltratively grow into the healthy brain parenchyma and are highly resistant toward most chemotherapeutic drugs and irradiation. Additionally, GBM cells shape an immunosuppressive microenvironment, which in turn leads to an escape of immune surveillance (for a review, see [2]). An upcoming approach to treat solid tumors such as GBM is oncolytic virotherapy (OVT) [3]. Oncolytic viruses (OV) are either original or genetically modified viruses that are able to replicate in tumor cells and ultimately kill these cells, whereas non-neoplastic cells are left unaffected (for a review, see [4]). However, to date, the outcomes of OVT are promising but scarce. In GBM, the efficiency of OVT is limited due to the infiltrative growth of this tumor. A few layers of tumor-ensheathing non-neoplastic cells will stop the virus spreading, thus outlying parts of the tumor will not be directly reached by OVs.

However, oncolytic adenoviruses (OAV) are able to induce immunogenic cell death (ICD), which is determined by the release of danger associated molecular pattern (DAMP) molecules ([5,6] and for a review, see [7]). This greatly affects the OV’s potential to induce an anti-tumor immune response and significantly contributes to their anti-tumoral effects [8]. Furthermore, OV infected cells release pathogen-associated molecular pattern (PAMP) molecules such as nucleic acids or virus proteins and induce the production of pro-inflammatory cytokines such as interferons (IFN; for a review, see [9]). Eventually, this leads to the attraction of dendritic cells (DC), promoting the uptake and presentation of tumor cell debris as well as of tumor specific neo-antigens by DCs, which finally primes anti-tumoral T cell responses [10]. On the other hand, GBM are highly immunosuppressive “cold” tumors. Immunosuppressive effects are maintained, aside from others, by the increased cell surface expression of immune checkpoint proteins (ICP) such as cytotoxic T-lymphocyte associated protein 4 (CTLA4) or programmed death ligand 1 (PD-L1) that counteract T-cell mediated tumor cell surveillance (for a review, see [11]). Immune-checkpoint inhibitors (ICI) that interrupt the interaction of ICPs and their ligands are effective in immunogenic “hot” tumors (for a review, see [12]). Only a small subset of GBM patients respond to ICIs, possibly because the PD-L1 mediated T-cell exhaustion is associated with the infiltration of M2-polarized, pro-tumorigenic macrophages [13,14]. Aside from the limited effects of ICI-treatment in GBM patients, their systemic application is associated with significant adverse events, elevated morbidity, and even mortality [15].

To treat GBM, it is important to use a suitable OV that (i) provides sufficient oncolytic capacity; (ii) initiates ICD; and (iii) stimulates anti-tumoral immune responses. In our study, we used the OAV XVir-N-31 [16], which has demonstrated therapeutic efficacy in preclinical tumor models [17,18,19]. Due to the deletion of the adenoviral E1A13S protein, the replication of XVir-N-31 depends on nuclear YB-1 expression, which is substantially upregulated in GBM [20]. We have upgraded XVir-N-31 to express a PD-L1 blocking antibody and analyzed the impact of XVir-N-31 and XVir-N-31-anti-PD-L1 on its immunostimulatory and therapeutic effects using immunohumanized, orthotopic mouse GBM models. We show that a single intratumoral injection of XVir-N-31 induced ICD and increased the number of tumor infiltrating lymphocytes (TILs). Both effects were further elevated if a blockade of the PD-1/PD-L1 axis was assembled. Based on our past and recent data, a clinical trial using XVir-N-31 to treat recurrent GBM patients was finally initiated and will start soon.

## 2. Results

### 2.1. Determination of Virus Mediated Cell Killing Activity and Virus Functionality

To compare the therapeutic effects, we first determined the cell killing activity of XVir-N-31 and XVir-N-31-anti-PD-L1 in the glioma cells and compared it to Ad-WT. As expected, Ad-WT showed high killing activity whereas both OAVs induced significantly lower cell death. Whilst the IC_50_ (48 h p.i.) of Ad-WT was around 10 MOI for LN-229 and less than 10 MOI for the U87MG cells, for XVir-N-31, it was 50 MOI in LN-229 and 30 MOI in the U87MG cells, and for XVir-N-31-anti-PD-L1 50 MOI in LN-229 and higher than 50 MOI in the U87MG cells (Figure 1).

To determine the function of anti-PD-L1 encoded by XVir-N-31-anti-PD-L1, the expression and secretion of anti-PD-L1 was detected as early as 48 h p.i. (Figure 2a). Secreted anti-PD-L1 was then tested to block the interaction of PD-L1 and PD-1. Indeed, the supernatants of the XVir-N-31-anti-PD-L1 infected cells demonstrated that anti-PD-L1 was functionally active (Figure 2b).

### 2.2. XVir-N-31 and XVir-N-31-Anti-PD-L1 Induce Immunogenic Cell Death

The induction of ICD in tumor cells by OVs is a hallmark of their therapeutic efficiency and is also necessary to induce a specific anti-tumoral immune response. As one sign of ICD is the release of DAMPs, we examined the virus-mediated release of HMGB1 and HSP70, the immunogenic protein YB-1 as well as the cell surface exposure of CRT. In vitro, XVir-N-31 and XVir-N-31-anti-PD-L1 induced HMGB1-, HSP70-, and YB-1-release and the surface exposure of CRT whilst Ad-WT showed very low or even no induction of DAMP release or CRT exposure. Interestingly, the surface expression of CRT was elevated in the XVir-N-31-anti-PD-L1 infected U87MG cells when compared to the infection of these cells using XVir-N-31. The PD-L1 status of GBM cells did not influence the induction of DAMP release (Figure 3a,b; Appendix A).

Using the U87MG (PD-L1^high^) immunohumanized mouse glioma model in vivo, the ICD was measured by staining for HMGB1 and HSP-70 using either OVTs alone or in combination with blocking the PD-1/PD-L1 interaction. We first stained for HMGB1 and HSP70 in the group of mice that received nivolumab monotherapy, which in a clinical trial failed to result in the improvement of the GBM patient’s overall survival (Checkmate-548 clinical trial, NCT02667587). We did not observe HSP70 staining in the tumors whereas HMGB1 staining was slightly detectable (Appendix A). Whilst in vitro the infection of GBM cells with Ad-WT led to the release of small amounts of HMGB1 and YB-1, no HSP70 release was observed (Figure 3a, Appendix A) and in vivo, no staining of HMGB1 and HSP70 was observed neither in the sham treated group of mice nor in the group that received an intratumoral injection of Ad-WT, and neither in the virus-injected nor in contralateral tumors. In contrast, HMGB1 and HSP70 staining was visible in the XVir-N-31, XVir-N-31 plus nivolumab and XVir-N-31-anti-PD-L1 treated animals. HMGB1 and HSP70 were not only apparent in the virus-injected ipsi-, but also in the contralateral tumors (Figure 3c,d; Appendix A). Interestingly, at least at the timepoint of analysis, no virus particles were detectable in the contralateral tumors as determined by staining for the adenovirus hexon capsid protein. In contrast, hexon staining was prominent in the virus-injected tumors (Figure 4). Combined treatment of the mice with XVir-N-31 plus nivolumab or XVir-N-31-anti-PD-L1 treatment further enhanced the amount of HMGB1 prominently in the virus-injected tumors. However, an elevated HMGB1 staining at these conditions was also visible in the contralateral tumors, indicating that both systemically applied nivolumab or intratumoral anti-PD-L1 expression further pushes ICD independent from the direct oncolytic effect of the viruses (Figure 1 and Figure 3c,d). Comparable results were observed for HSP70 (Appendix A).

### 2.3. Virus Mediated IFNγ Release

To determine whether the virus infection of glioma cells subsequently induced an immunostimulatory boost aside from the induction of ICD, we measured the release of IFNγ in cocultures of virus-infected glioma cells and HLA-matched PBMCs, the latter being unaffected by the infection with XVir-N-31 (Appendix A). No elevated IFNγ level was observed in the cocultures of PBMCs with the Ad-NULL-infected GBM cells. In contrast, the IFNγ concentrations increased after lytic adenovirus infection, independently of whether the virus was an OAV (XVir-N-31, XVir-N-31-anti-PD-L1) or not (Ad-WT), or whether DAMPs had been released. Remarkably, when infecting PD-L1-positive U87MG cells with XVir-N-31-anti-PD-L1, compared to XVir-N-31 infection, the IFNγ concentrations increased further whereas this was not the case in the PD-L1-negative LN-229 cells (Figure 5), indicating an additional effect of anti-PD-L1 in IFNγ production by either the glioma cells or by co-cultured lymphocytes.

### 2.4. XVir-N-31 Leads to an Elevated Number of TILs That Can Be Further Raised by Immune Checkpoint Inhibition

It has been shown for OVs that oncolysis often induces lymphocyte infiltration into the tumor [21,22]. As OV mediated therapeutic effects can be fostered by the treatment with ICIs [21,23,24,25], we were interested in whether an additional blockade of the PD-1/PD-L1 interaction further enhanced the number of TILs. For this, in one group of mice, we intratumorally injected XVir-N-31 and systemically applied nivolumab, whilst another group received a single intratumoral injection of XVir-N-31-anti-PD-L1. While no CD45^+^ cells were detected in the healthy brain adjacent to the tumor (Appendix A), very few CD45^+^ cells were found in the tumor-bearing animals that received the sham treatment or nivolumab only (Figure 6a,b, Appendix A). Ad-WT injection led to a significant, approximately two-fold enrichment of human CD45^+^ TILs in the ipsi- and contralateral tumors. Compared to Ad-WT, the number of human CD45^+^ TILs was highly elevated by XVir-N-31 (Figure 6a,b, Appendix A). The combination of XVir-N-31 plus Nivolumab, or XVir-N-31-anti-PD-L1 further enhanced the human CD45^+^ TIL numbers.

We next analyzed the subtype of TILs by staining for the total (CD3^+^), cytotoxic (CD8^+^), helper (CD4^+^), and regulatory T cells (T_reg_; CD4^+^/FoxP3^+^) as well as for the NK cells (CD45^+^/CD56^+^). In the Ad-WT treated animals, no significant increase in the CD8^+^ cells was observed, while in the OAV treated mice, the number of CD3^+^/CD8^+^ cells increased about 20-fold in the ipsi- and contralateral tumors and was further elevated if the mice received nivolumab in addition, or were tumor-injected with XVir-N-31-anti-PD-L1 (Figure 6c,d and Appendix A). The CD4^+^ helper TILs were elevated by all viruses. However, the number of CD4^+^ TILs was significantly higher in the XVir-N-31 and XVir-N-31-anti-PD-L1 treated animals and was further elevated by blocking the PD-1/PD-L1 interaction (Figure 6e, Appendix A). In accordance with the immuno-suppressive capability of Ad-WT, human CD3^+^ TILs in mice that received Ad-WT did not express the T-cell activation marker CD134/OX40, whilst it was expressed in nearly all human CD3^+^ TILs (both in injected as well as in contralateral tumors) of XVir-N-31 and XVir-N-31-anti-PD-L1 treated mice (Figure 6f, Appendix A). Due to the known immunosuppressive activity of adenoviruses, we also analyzed the amount of CD3^+^/FoxP3^+^ T_regs_ in the ipsi- and contralateral tumors. Even if T_reg_ cells were detected in the tumors of the virus treated mice, the number of T_regs_ was much higher in the Ad-WT compared to XVir-N-31 and XVir-N-31-anti-PD-L1 treated mice (Figure 6g, Appendix A). The number of NK cells was elevated by both OAVs but not by Ad-WT. However, this effect was not persistently further elevated by blocking the PD-1/PD-L1 interaction (Figure 6h, Appendix A). The total number and the increased infiltration of human CD3^+^ TILs compared to NK cells indicate that OAVs mainly induced an adaptive, and only to a lesser amount an innate anti-tumoral immune response, at least in our animal model.

Unsurprisingly, PD-L1 has to be expressed on GBM cells to evoke a stimulatory effect by blocking the PD-1/PD-L1 interaction since in mice harboring PD-L1 negative LN-229 tumors, no additional stimulatory effect with regard to the amount of TILs was observed by XVir-N-31-anti-PD-L1 (Appendix A).

### 2.5. XVir-N-31-Anti-PD-L1 or XVir-N-31 in Combination with Nivolumab Presented an Abscopal Effect in the Reduction in Tumor Growth

Finally, we determined the tumor growth of the virus-injected and contralateral U87MG tumors. The sham treated mice developed large tumors in both hemispheres (Figure 7). No tumor growth reduction was observed by nivolumab alone (Appendix A). In contrast, virus injected tumors showed a massive growth reduction independently if Ad-WT or OAVs had been injected or whether nivolumab was systemically applied in addition to XVir-N-31. This indicates that if viruses are directly injected into the tumor, the lytic effect of these viruses is sufficient to kill most of the tumor cells and reduce the tumor volume. However, the situation in the contralateral tumors was completely different. Whereas Ad-WT and XVir-N-31 showed no effect on the tumor growth, the combination of XVir-N-31 plus nivolumab or XVir-N-31-anti-PD-L1 treatment resulted in a significant tumor volume reduction (Figure 7).

## 3. Discussion

Despite the successful development of OAVs and their use in clinical trials, a comprehensive description of the cell death induced by these viruses and their contribution to elicit an anti-tumor immune response is not fully understood. Therefore, when using an OAV to treat cancer, it is important to consider the characteristics of the virus as well as of the cancer cells, since these details affect the viral replication, oncolysis, and the resulting mode of cell death. The kind of cell death initiated after viral infection can dictate anti-tumor immune responses. It has been previously shown that the induction of ICD by OVs is a key issue to prime and activate anti-tumor immune responses [26]. In our study, we used the YB-1 dependent OAV XVir-N-31 to treat GBM [16]. XVir-N-31 due to YB-1 expression in glioma cells, especially in high grade glioma, efficiently replicates, induces oncolysis, and kills these cells, although a significant higher MOI is required compared to Ad-WT (Figure 1; [27]).

To determine the immunostimulating effects of the XVir-N-31 based OVT in vivo, we used an immunohumanized, orthotopic NSG mouse GBM model. One limitation of this mouse model is the development of graft-versus-host disease (GvHD) over time, which makes survival analyses impossible. To avoid effects induced by GvHD, all mice were sacrificed and tumors were analyzed long before GvHD was observed (Appendix A). Since NSG mice did not harbor functional T, B, and NK cells that also cannot be activated and so will not be functionally involved in immunostimulatory processes, all analyses regarding TIL infiltration in the virus injected as well as in the contralateral tumors were only performed for human immune cells.

In glioma, the infection with XVir-N-31 or its offspring XVir-N-31-anti-PD-L1 leads to the induction of ICD in vitro and in vivo, as indicated by the release of the DAMPs HMGB1 and HSP70 or by the surface exposure of the “eat me” protein CRT whereas Ad-WT, expressing E1A13S and E319K proteins that are not present in XVir-N-31 and XVir-N-31-anti-PD-L1 and that are known to possess immunosuppressive functions and counterattack immune recognition, did not [19,28,29]. Interestingly, the surface expression of CRT was elevated in the XVir-N-31-anti-PD-L1 infected U87MG cells when compared to the infection of these cells using XVir-N-31. Whether this elevated CRT level may contribute to anti-tumor immunity needs further investigation. Moreover, it would certainly also be of interest to clarify the molecular background of this phenomenon as well as the involvement of CD47, which is known to counterbalance the pro-phagocytic CRT signal [30].

However, the absence of the T-cell activation marker CD134 in the Ad-WT infected tumors was striking and indicates that it is not required that XVir-N-31 has to be armed with CD134/OX40 to drive the CD8^+^ T-cell responses [31]. Although we do not know the molecular basis for this striking effect of XVir-N-31 on CD134 activation, it might be explained by the inhibitory function of the viral large E1A13S protein and its impact on histon acetylation patterns that regulate gene expression [32,33,34,35]. Current research is addressing this very important issue. Additionally, it has been reported that the adenoviral E1B19K protein (also not expressed in XVir-N-31) limits the local host’s innate immune inflammation [36]. The release of YB-1 was also noticeable after infecting the GBM cells with our OAVs. YB-1 is a factor that is involved in inflammatory responses [37] and has been identified as a tumor-associated antigen capable of eliciting a T-cell response [6]. However, the designation of YB-1 as a proper DAMP protein needs further investigation.

An interesting finding from our subtype analysis of TILs was also the increased number of T_regs_ in the tumor area of Ad-WT compared to the XVir-N-31 and XVir-N-31-anti-PD-L1 treated mice. It has recently been shown that CD134/OX40 costimulation abrogates the suppressive function of T_regs_ [38]. Thus, the activation of CD134 by XVir-N-31 might, in addition, indirectly limit the T_reg_ mediated suppression in tumor immunity [38]. However, elucidating the contributions of adenoviral proteins in these processes requires detailed studies, which was not the aim of this study. Nevertheless, our data suggest that the expression of the adenovirus E1A13S and E1B19K proteins present in Ad-WT should be avoided due to their immunosuppressive effects, thereby accepting the loss of potency. Our findings are in line with the results obtained for the Herpes simplex virus, demonstrating that the induction of an anti-tumor immune response to treat cancer with OVs is more important than efficient virus replication [26,39].

Notably, the induction of ICD was not exclusive for OAV injected tumors, but was also present in the contralateral tumors, mimicking infiltratively growing GM cells (Figure 3, Appendix A). In colon cancer, OAVs can be transferred to metastases located far away from virus injected tumors via tumor-derived exosomes, in this way inducing abscopal effects due to the oncolysis of metastatic cells by OAVs transported by these exosomes [40]. However, at least in the late stages (day 35), we did not observe any virus replication in the contralateral gliomas (Figure 4) indicating that, at least for brain tumors, the growth reduction we saw for contralateral tumors did not seem to be a result of oncolysis, but might be explained by a kind of “secondary ICD burst“ induced by CD8^+^ cytotoxic TILs [41]. This is in line with our observation that CD8^+^ cytotoxic TILs were highly enriched in the virus-injected, but also in the uninjected tumors (Figure 6, Appendix A).

In addition, viral infection induced a general immmunostimulatory boost via the release of IFNγ. However, the amount of IFNγ was nearly equal for Ad-WT, XVir-N-31, and XVir-N-31-anti-PD-L1 (Figure 5), suggesting that this was a result of virus infection and not limited to OAVs. As we used cocultures of OAV infected GBM cells and HLA-A/B matched PBMCs (to mimic the situation in the tumor area) to determine the IFNγ release, we could not discriminate whether the IFNγ we measured originated from the OAV infected GBM cells or from the cocultured PBMCs or from both. However, in vivo, the origin of IFNγ in the tumor tissue should not make any difference in the induction of general immunostimulatory processes. Probably as a result of the induction of ICD and proinflammatory IFNγ, the XVir-N-31 injected tumors showed an increased infiltration of activated CD4^+^ and CD8^+^ T lymphocytes (Figure 6) and a massive reduction in the tumor growth (Figure 7). As in our study, PBMCs of only one donor were used to engraft the mice, one should bear in mind that dependent on the PBMC donor, there might be a variability in the induction of immunostimulatory effects by our OVT.

As a limitation of XVir-N-31, no growth reduction was observed in the contralateral tumors, although activated CD8^+^ T cells were enriched (Figure 6). However, the cell count was significantly lower compared to the XVir-N-31-anti-PD-L1 treated animals and the amount of HMGB1 in these tumors was also low. We hypothesize that an OAV independent, second burst of immunostimulation by an efficient induction of ICD is necessary to note a therapeutically relevant abscopal effect for an XVir-N-31 based immunotherapy. Nevertheless, we believe that XVir-N-31 fulfills the criteria as an oncolytic and immunostimulating agent for the treatment of GBM.

In contrast to “hot” melanoma, which often has been used to demonstrate the therapeutic effects of OVT, GBM are “cold”, immunosuppressive tumors (for review, see [42]). One cause of its immunosurveillance escape is the expression of PD-L1. PD-L1 is expressed on the majority, especially on the most malignant mesenchymal subtype of GBM, and its expression is associated with further immunosuppressive effects such as infiltration and M2 polarization of the tumor associated macrophages (TAMs) [14,43]. Up until now, several preclinical trials have demonstrated immunostimulatory and even abscopal effects of OVs if used in combination with ICIs. These trials also indicate that a combined OV/ICI therapy is feasible but has to be further optimized [23,44,45,46,47,48]. We therefore examined whether an ICI therapy, combined with our OAVs, provides a therapeutic benefit. In this regard, we armed XVir-N-31 to express anti-PD-L1. In parallel, we used XVir-N-31 in combination with repeated systemic injections of nivolumab as it is practiced in clinical trials. We selected nivolumab instead of atezolizumab, since this seemed to be the strongest contestant in comparison to the XVir-N-31-anti-PD-L1 expressed anti-PD-L1 as PD-1 expressed on lymphocytes should already be capped by nivolumab outside the brain, thus avoiding the exhaustion of TILs by GBM expressed PD-L1 during the infiltration phase. In contrast, the virus-expressed anti-PD-L1 antibody covers PD-L1 on the tumor cells, protecting TILs from exhaustion during tumor infiltration. Whilst nivolumab monotherapy did not induce DAMP release in the U87MG tumors nor elevate the number of TILs (Appendix A and Appendix A), an additional ICI resulted in a further elevated induction of ICD and a higher number of TILs not only in the OAV-injected, but also in the contralateral tumors (Figure 3, Appendix A). This indicates that the elevated DAMP release in contralateral tumors by a combined OAV/ICI treatment might be evoked by cytotoxic T-cell mediated GBM cell death. In accordance, the growth of contralateral tumors was significantly reduced by either additional nivolumab treatment or by XVir-N-31-anti-PD-L1. ICIs, if systemically applied, can lead to severe adverse events in several organs as well as elevated morbidity and mortality [15]. The intratumoral injection of XVir-N-31-anti-PD-L1 might restrict anti-PD-L1 availability to the tumor area adjacent to the site of virus injection. This might minimize the known severe adverse effects observed in cancer patients treated with atezolizumab or nivolumab and opens up future potential to use XVir-N-31 as a carrier for inhibitors that target other immune checkpoint proteins.

Until now, it is not known whether PD-L1 negative or low expressing GBM XVir-N-31-anti-PD-L1 instead of XVir-N-31 might provide a therapeutic benefit. However, on one hand, XVir-N-31-anti-PD-L1, at least in vitro, was superior to XVir-N-31 in the induction of CRT surface exposure in the OAV-infected GBM cells (Figure 3b). On the other hand, in mice harboring PD-L1 negative LN-229 GBM, we did not observe differences in the infiltration of immune cells into the tumor area, as was observed for mice harboring PD-L1^high^ U87MG tumors. Therefore, in the future, if planning OVT for GBM patients, it might be an advantage to determine the PD-L1 status of the tumor tissue.

## 4. Material and Methods

### 4.1. Cells, Cell Lines, and Cell Culture

HEK293 cells were from Microbix (Mississauga, ON, Canada) and the human GBM cell lines U87MG and LN-229 were from N. Tribolet (Geneva, Switzerland). The PD-L1 status of the U87MG (PD-L1 high) and LN-229 (PD-L1 negative) cells was determined by FACS analysis (Appendix A). Cells were maintained in Dulbecco’s modified Eagle’s medium (DMEM), 10% fetal calf serum (FCS), and 1% penicillin-streptomycin (P/S, Sigma-Aldrich, Darmstadt, Germany). HLA-A/B phenotyping of the U87MG and LN-229 cells has been routinely conducted at the Institute for Immunopathology (University Hospital Tübingen, UKT, Germany). Both cell lines showed the phenotype A*02:01, B*13:02, *27:05; C*06:02, *01:02; DRB1*07:01, *11:03; and DQB1*02:02, *03:01. Human HLA A/B-matched peripheral blood mononuclear cells (PBMC), obtained from the Transfusion Medicine Department (UKT, Germany), were isolated from buffy coats by Ficoll–Plaque PLUS gradients (GE Healthcare, Chalfont St Giles, UK) and were cultured in RPMI medium 1640, 10% FCS, 1% P/S, 1% non-essential amino acids (Sigma-Aldrich), or stored frozen for further use. All cells were cultured at 37 °C in a humidified, 5% CO_2_-containing atmosphere.

### 4.2. Adenoviral Vectors and Infection

Wildtype adenovirus (Ad-WT) substrain dl309 and XVir-N-31 have been described [16,49]. Replication deficient Ad-NULL lacks the E1 gene (SignaGen Laboratories, Frederick, MA, USA). XVir-N-31-anti-PD-L1 was generated using the AdEasy System (Agilent Technologies, Waldbronn, Germany). The anti-PD-L1 transgene is composed of a domain encoding the antigen binding Fab fragment, which consists of the variable light chain, the Gly4Ser3 linker, and the variable heavy chain and a domain encoding for the Fc fragment of the antibody. At the 5′ end, the transgene contains a Kozak sequence, the start codon, an IgK leader sequence, and a HA tag. The Ig kappa leader was used to enhance the secretion of the protein. Sequences of the variable light and heavy chains connected by the Gly4Ser3 linker originated from the patent US20100203056 A1. The cassette was inserted into the E3-region of XVir-N-31 via *DraI* (nt28706 and nt29308 with respect to AY339865.1), replacing the E319K protein. A stop codon in the hinge region resulted in the expression of a human PD-L1 specific single chain antibody (anti-PD-L1). The virus genome was sequenced to validate the correctness of cloning. The exact extensive cloning strategy can be provided on request. All viruses were prepared, purified, and titrated as described [27].

### 4.3. Cell Viability Assays

For the detection of OAV mediated cell killing (oncolysis), cells were infected with different MOI of the indicated virus. A total of 48 h after infection, photographs were taken using an Axiovert 200M microscope (Zeiss, Wetzlar, Germany). Cell viability was measured using a MTT Kit (Merck, Darmstadt, Germany). Absorbance was determined on a MULTISKAN EX reader (Thermo Electron, Langenselbold, Germany).

### 4.4. Determination of Virus Copy Numbers

Genomic virus DNA was isolated using the DNeasy Blood and Tissue Kit (Qiagen, Hilden, Germany). A total of 20 ng of DNA was analyzed using the SYBR green PCR Mastermix I (Eurogentech, Liege, Belgium) on a C1000™ Thermal Cycler CFX96™ (Bio-Rad, Hercules, CA, USA) at the following conditions (95 °C 2 min, 45 cycles of 94 °C 15 s, 60 °C 15 s, 72 °C 15 s) with hexon (Hexon-frwd GGCCATTACCTTTGACTCTTC; Hexon-rev GCATTTGTACCAGGAACCAGTC) or the GAPDH specific primer (GAPDH-frwd TGGCATGGACTGTGGTCATGAG, GAPDH-rev ACTGGCGTGTTCACCACCATGG). Quantification of the virus copy numbers was determined using the ∆∆CT equation [50].

### 4.5. Immunoblot Analyses

To determine the expression of the HA-tagged PD-L1 antibody coded by XVir-N-31-anti-PD-L1, 48 h after infection (30 MOI), the cell lysates were prepared. For the detection of anti-PD-L1 secretion, cells were infected as described above. One hour later, the medium was changed to a serum-deprived medium. The supernatants were collected after 48 or 72 h, and the proteins were acetone precipitated and prepared for immunoblotting using the following antibodies: HA-tag (1:500, No. 14-6756-81, Thermo Fisher Scientific) and GAPDH (1:1000, sc-25778, Santa Cruz, Heidelberg, Germany). The detection was performed using Clarity ECL substrates on a ChemiDoc^TM^ MP Imaging System and ImageLab 5.1 software (all Bio-Rad Laboratories, Feldkirchen, Germany).

### 4.6. PD-1/PD-L1 Binding Assay

The supernatants from the OAV-infected or control cells were collected, concentrated using Centriplus Centrifugal filter devices (Merck-Millipore, Darmstadt, Germany), and used in a PD-1/PD-L1 Blockade Assay Kit (Promega, Walldorf, Germany). Luminescence was measured on a TriStar2 S LB 942 Multimode Reader (Berthold Technologies, Bad Wildbad, Germany).

### 4.7. ELISA Based Detection of HMGB1, HSP70, YB-1 and IFNγ

The release of HSP70, HMGB1, and of YB-1 was examined by ELISA (Hölzel Diagnostics, Köln, Germany). Supernatants of the infected cells were collected at the time point of 50% oncolysis, cleared by centrifugation and used in the respective ELISA kit. For the detection of IFNγ released from virus infected GBM cells and/or from tumor infiltrating lymphocytes, we performed an in vitro coculture model where we first infected GBM cells with OAVs. The virus-infected GBM cells were intensively washed 4 h after infection with PBS to remove the residual virus particles, cultured for another 20 h, and were subsequently cocultured with HLA-matched, interleukin-2 stimulated PBMCs (IL-2; 100 IU/mL, Immuno Tools, Friesoythe, Germany). After a period of 48 h, the supernatants were collected and the IFNγ was measured using the MAX™ Deluxe Human IFN-γ ELISA (Biolegend/Biozol, Eching, Germany).

### 4.8. Flow Cytometry Analysis (FACS)

The PD-L1 expression was determined by FACS using a PD-L1 specific (Thermo Fisher, #53-5983-42) or a mouse IgG1κ isotype control antibody (Thermo Fisher, #53-4714-80). For the surface exposure of calreticulin (CRT), infected glioma cells at 50% oncolysis were collected and stained with a CRT specific antibody (Novus Bio #B-4-120621). For human immune cell engraftment in mice, blood was taken, mixed 1:1 with 10 μM EDTA, and incubated with both the FITC-anti-mouse CD45 (Biolegend, Eching, Germany, #103108) and APC-anti-human CD45 (Biolegend, #304012) antibodies. After washing, the erythrocytes were removed by ACK buffer lysis (Thermo Fisher). Analyses were performed on a MACSQuant Analyzer 10 Flow Cytometer (Miltenyi Biotec, Bergisch-Gladbach, Germany), and data were analyzed using FlowJo v10 (Ashland, OR, USA).

### 4.9. Immuno-Humanized Mouse Model

Immuno-humanized mice are a promising translational model for studying human immunity. Mouse strains such as NOD-Scid IL-2 receptor gamma (null) mice (short name: NSG) provide a deleted interleukin (IL)-2 receptor common gamma chain, leading to the lack of adaptive immune functions. Furthermore, these mice displayed multiple defects in innate immunity, support elevated levels of human hematolymphoid engraftment, and allow for the growth of human tumors. Therefore, immunohumanized mice support studies in many areas of immunology and cancer [51,52]. Animal work was performed in accordance with the German Animal Welfare Act and its guidelines (e.g., 3R principle) and was approved by the regional council of Tübingen (approval N02/19G). NOD.Cg-*Prkdc^scid^ Il2rg^tm1Wjl^*/SzJ (NSG) mice (Jackson Laboratory, Bar Harbor, ME, USA) were bred in IVC cages in the animal facility of the institute under sterile conditions and used at an age of 2–6 months. Using anesthesia and analgesia, 1 × 10^5^ U87MG or LN-229 cells were stereotactically implanted into both striata, and the mice were intensively monitored to avoid and reduce pain. Ten days later, 2 × 10^6^ human PBMCs (HLA-A/B identical to GBM cells) were injected into the tail vein. After engraftment, the mice received the antibiotic Cotrim E (230 mg/L, Ratiopharm, Ulm, Germany) in their drinking water. The immune cell engraftment was supervised by the detection of human CD45^+^ cells in the blood of mice, as described above. After engraftment, the mice were randomly split into the treatment groups, being sex and age, equally distributed as best as possible. PBS (sham) or 3 × 10^8^ infectious particles (IFU) of the indicated viruses were injected into the tumor located in the right hemisphere. The contralaterally growing tumor, which should mimic the brain parenchyma infiltrating GBM cells located far away from the virus injection side, was left untreated. Mice of the respective groups were treated intraperitoneally with nivolumab (200 μg) on days 4, 7, 10, and 14 after virus injection (Appendix A).

One disadvantage of this mouse model is the development of graft-versus-host disease (GvHD) over time, which makes survival analyses impossible. To avoid effects induced by GvHD, we determined the timepoint of its onset, which becomes visible by a sudden massive burst of human CD45^+^ cells in the blood and by animal weight loss. We observed this burst around day 80 after engraftment, slightly later than that described by Ehx et al. [53]. At this time point, the mice did not lose body weight or show any other visible signs of GvHD such as scrubby fur (Appendix A).

As the median survival of mice bearing U87MG or LN-229 tumors is around 40 days [54,55], we sacrificed, according to the legal rules, all mice at day 35 after treatment, which was also the time point the first mouse developed neurological symptoms associated with tumor growth. Mouse brains were collected and prepared for further analyses.

### 4.10. Histology and Immunofluorescence

Brains were fixed in 4% PFA, dehydrated in 20% and 30% sucrose and cryosectioned using a Leica Cryomicrotome CM3050S (Leica Mikrosystems GmbH, Wetzlar, Germany). Tissue sections were PBS-washed, pretreated for antigen retrieval by boiling in Tris-EDTA buffer pH 9 (15 min), and washed and treated with 3% H_2_O_2_/methanol (10 min) to block peroxidase activity. After PBS washing, sections were blocked (3% animal serum) and the tissue was stained using the following antibodies: human nuclei (Millipore, MAB1281C3), each human CD45, CD3, CD4, CD8, FoxP3 (all Invitrogen, Walham, MA, USA, #14-0459-82; #14-0038-82; #14-0459-82; #14-0089-82; #14-4777-80;), CD56 (Biolegend, #304602;) and CD134 (eBioscience, San Diego, CA, USA; #14-1347-82), Hexon (Santa Cruz Biotechnology, #F0517), HMGB1 (Invitrogen, MA5-17278), or HSP70 (Invitrogen, # MA3-007). After washing, fluorochrome conjugated secondary antibodies were added (Invitrogen, #2208228; VC295507), slices were washed again and mounted with Vectashield-DAPI mounting media (Biozol). Images were taken on a Zeiss LSM 710 confocal microscope. TILs were counted using ImageJ (Rasband, NIH, Bethesda, MD, USA).

### 4.11. Hematoxylin and Eosin (H&E) Staining and Tumor Volumetry

Slides of mouse brain tissue were generated as described and were incubated with 0.1% hematoxylin (10 min), stained with 0.5% eosin/methanol solution (90 s, both: Sigma-Aldrich), and incubated under running tap water. Subsequent dehydration using an alcohol dilution series was followed by Permount mounting (Fisher Chemical; #202282). To determine the tumor size, the start and end of the tumors were determined and the area of the tumor was measured every 100 μm using ImageJ. The surface area multiplied by the thickness of the section (until the next section) gave the partial volume. The sum of all partial volumes approximated the whole tumor volume.

### 4.12. Statistical Analysis

All in vitro experiments were performed at least thrice if not mentioned otherwise. For the in vivo experiments, the group and sample size are indicated in the figure legends. To assume a Gaussian distribution, all data received from the in vitro experiments passed a normality test (Shapiro–Wilk and Tukey’s multiple comparison tests). Further statistical analyses were conducted with a two-tailed Student’s t-test or one-way ANOVA using GraphPad Prism 7.0 (GraphPad Inc., San Diego, CA, USA). The results are represented as the mean ± standard error mean (SEM). The *p*-values of < 0.05 were considered as statistically significant (n.s.: not significant; * *p* < 0.05; ** *p* < 0.01; *** *p* < 0.001; **** *p* < 0.0001). All obtained data were stored online under www.figshare.com under the following accession number (https://doi.org/10.6084/m9.figshare.c.6174568, accessed on 28 August 2022).

## 5. Conclusions

Taken together, our data revealed a much stronger anti-tumor immunity of XVir-N-31 in GBM than Ad-WT, although XVir-N-31 demonstrated a significant lower cell killing capacity compared to Ad-WT. The strong anti-tumor immunity induced by our OAVs is characterized by their efficient immunostimulating properties including the induction of ICD, which was not observed for Ad-WT. Moreover, our data also strongly indicate that additional ICI based therapy is required to achieve a significant tumor reduction and abscopal effects in virus-untreated tumor sites, thus improving the therapeutic outcome. Finally, based on our data, it seems reasonable to suppose that using an improved “tumor-immunostimulating” OV in conjunction with immune therapy has the effectiveness to become a clinical reality. Based on our data from this and former studies, a phase I study using XVir-N-31 to treat recurrent GBM is scheduled to start in 2023.

## Figures and Tables

**Figure 1 ijms-23-09965-f001:**
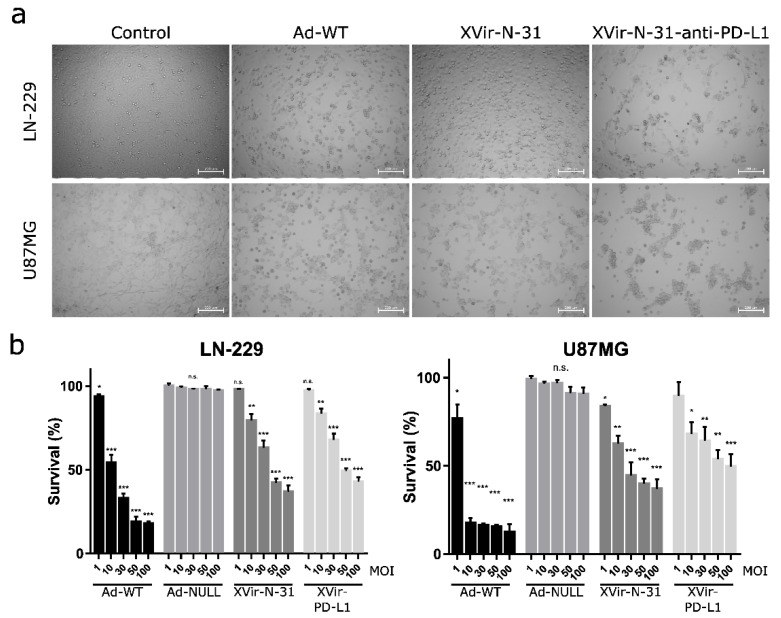
Cell killing activity of OAV XVir-N-31 and XVir-N-31-anti-PD-L1. (**a**) LN-229 or U87MG cells were infected with 50 MOI of Ad-WT, XVir-N-31, XVir-N-31-anti-PD-L1, or were left untreated (control). Pictures were taken 48 h later. Dead or dying cells could be identified by their rounded shape. (**b**) LN-229 or U87MG cells were infected with increasing MOI of the indicated viruses. OAV mediated cell killing was measured 48 h after infection using the MTT assay (XVir-PD-L1: XVir-N-31-anti-PD-L1; *n* = 3; SEM; n.s.: not significant; * *p* < 0.05, ** *p* < 0.01, *** *p* < 0.001).

**Figure 2 ijms-23-09965-f002:**
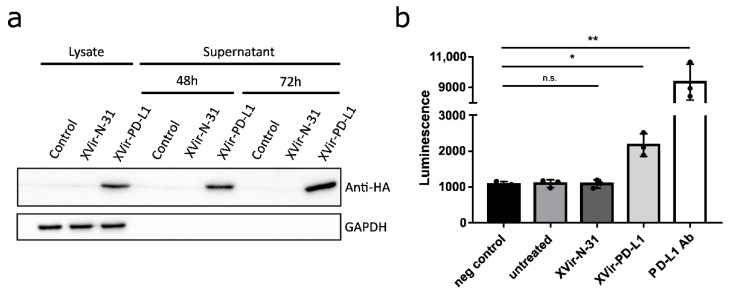
XVir-N-31-anti-PD-L1 expresses a functional PD-L1 antibody. (**a**) HEK293 cells were infected with 30 MOI of XVir-N-31 or XVir-N-31-anti-PD-L1, or were left untreated (control). Lysates or supernatants were collected at the indicated time points. The production and secretion of anti-PD-L1 were analyzed by immunoblot using an anti-HA antibody, GAPDH served as the loading control. (**b**) HEK293 cells were infected with 50 MOI of XVir-N-31, XVir-N-31-anti-PD-L1, or were left untreated. The supernatants were collected 48 h after infection and were subsequently used in an PD1/PD-L1 blocking assay. Increased luminescence indicates the PD-1/PD-L1 blockade. PBS served as the negative control, 10 μg/mL anti-PD-L1 antibody for the positive control (XVir-PD-L1: XVir-N-31-anti-PD-L1; n = 3; SEM; n.s.: not significant; * *p* < 0.05, ** *p* < 0.01).

**Figure 3 ijms-23-09965-f003:**
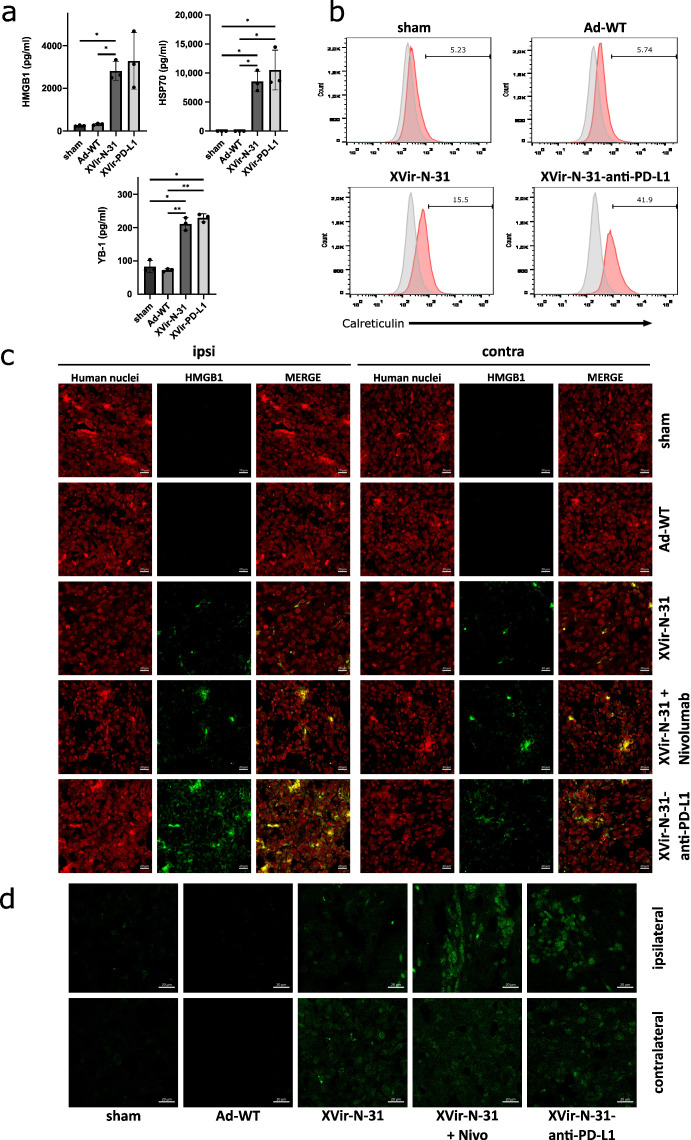
Induction of immunogenic cell death by XVir-N-31 and XVir-N-31-anti-PD-L1. (**a**) U87MG cells were infected with 50 MOI XVir-N-31 or XVir-N-31-anti-PD-L1, or with 20 MOI Ad-WT, or were left untreated. Supernatants were taken at the timepoint the cultures showed 50% oncolysis, and were analyzed for HMGB1, HSP70, or YB-1 release by ELISA (XVir-PD-L1: XVir-N-31-anti-PD-L1). (**b**) CRT surface expression was analyzed in the U87MG cells at the same conditions as indicated in A (n = 3; SEM; * *p* < 0.05; ** *p* < 0.01). (**c**) The detection of HMGB1 in U87MG ipsilateral virus-injected and contralateral tumors 35 days after intratumoral injection of either PBS (sham), 3 × 10^8^ IFU of either Ad-WT, of XVir-N-31 alone or in combination with multiple systemic applications of nivolumab, or of XVir-N-31-anti-PD-L1 (n = 7–8 mice per group; representative pictures are shown). (**d**) Enlightenment of the HMGB1 staining as indicated in (**c**).

**Figure 4 ijms-23-09965-f004:**
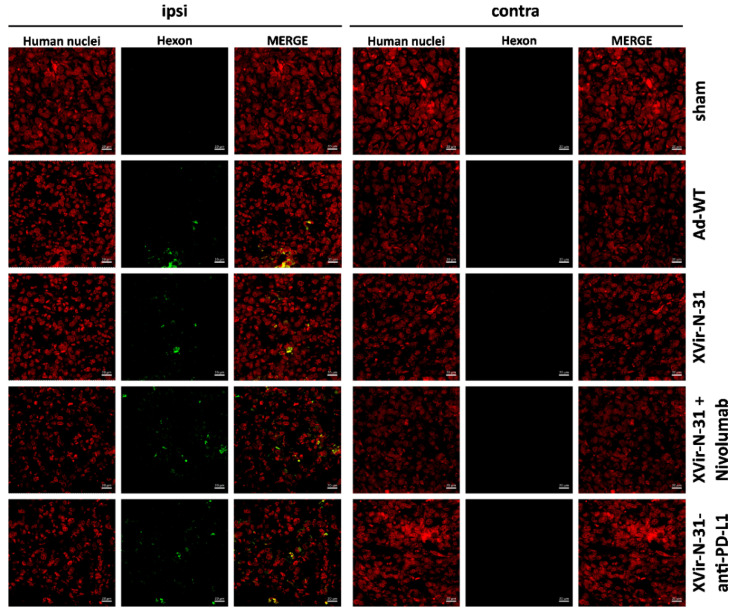
Hexon staining of ipsilateral injected and contralateral non-treated tumors. The detection of the adenoviral hexon protein (green) in the U87MG ipsilateral virus-injected and contralateral tumors (identified by staining for human nuclei, red) 35 days after intratumoral injection of either PBS (sham), 3 × 10^8^ IFU of either Ad-WT, of XVir-N-31 alone or in combination with multiple systemic applications of nivolumab, or of XVir-N-31-anti-PD-L1 (n = 7–8 mice per group; representative pictures are shown).

**Figure 5 ijms-23-09965-f005:**
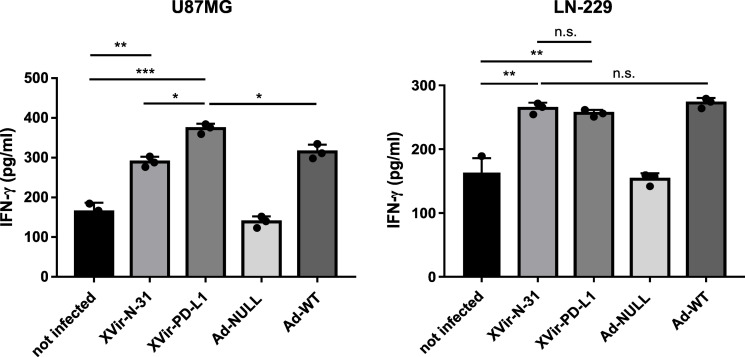
IFN-γ release from cocultures of HLA-matched human PBMCs with virus infected LN-229 or U87MG cells was measured by ELISA. The cells were infected with 30 MOI, were intensively washed with PBS 4 h later, collected 24 h p.i. and were then cocultured with HLA-matched PBMCs that had been treated with IL-2 (100 IU/mL, Immuno Tools, Friesoythe, Germany) 6 days before use. A total of 48 h later, the supernatants were collected and IFN-γ was measured by ELISA (XVir-PD-L1: XVir-N-31-anti-PD-L1; n = 3; SEM; n.s.: not significant; * *p* < 0.05, ** *p* < 0.01, *** *p* < 0.001).

**Figure 6 ijms-23-09965-f006:**
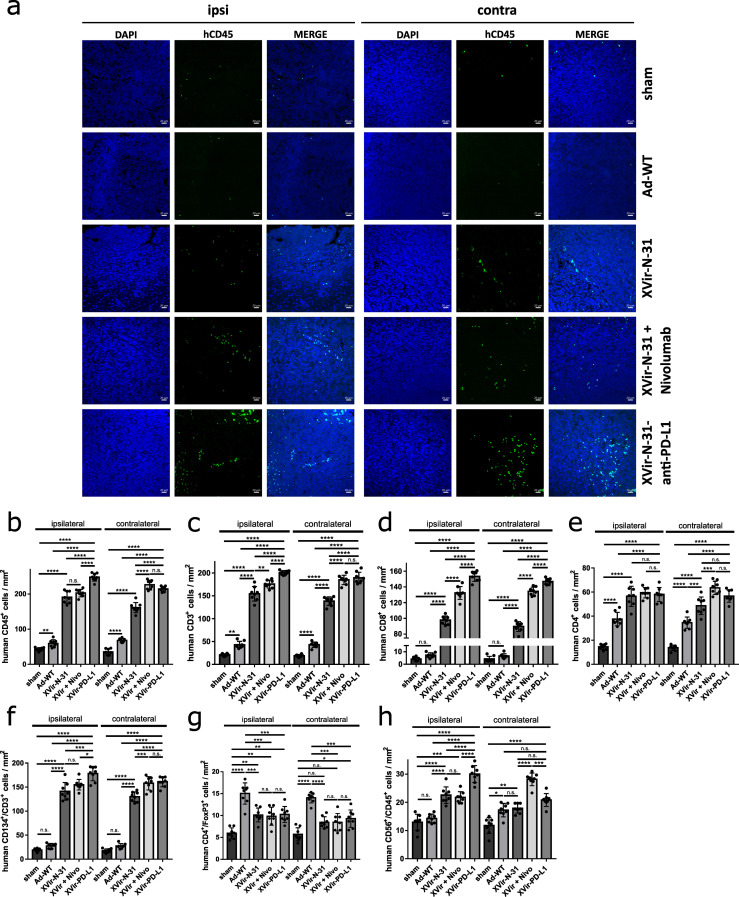
Intratumoral immune cell invasion after OVT. (**a**) Representative immunofluorescence pictures of human immune cell infiltration into the U87MG tumors in NSG mice (n = 8 mice per group, five sections per tumor, per side). Brain sections were stained with DAPI (blue) and anti-human CD45 (green). Pictures were taken 35 days after the injection of the sham virus into the tumor on the ipsilateral (injected) as well as on the contralateral (not injected) side. (**b**–**h**) Quantification of the infiltration of different immune cells per mm^2^ in the GBMs located in the ipsilateral or contralateral hemisphere ((**b**) human CD45^+^ cells; (**c**) human CD3^+^ T cells; (**d**) human CD8^+^ cytotoxic T cells; (**e**) human CD4^+^ T helper cells; (**f**) human CD3^+^ T cells expressing the activation marker CD134; (**g**) human CD4^+^/FoxP3^+^ regulatory T cells; (**h**) human CD68^+^/CD45^+^ NK cells). (XVir-PD-L1: XVir-N-31-anti-PD-L1; n = 7–8 tumors and five slices per were analyzed; SEM; n.s.: not significant; * *p* < 0.05; ** *p* < 0.01, *** *p* < 0.001, **** *p* < 0.0001).

**Figure 7 ijms-23-09965-f007:**
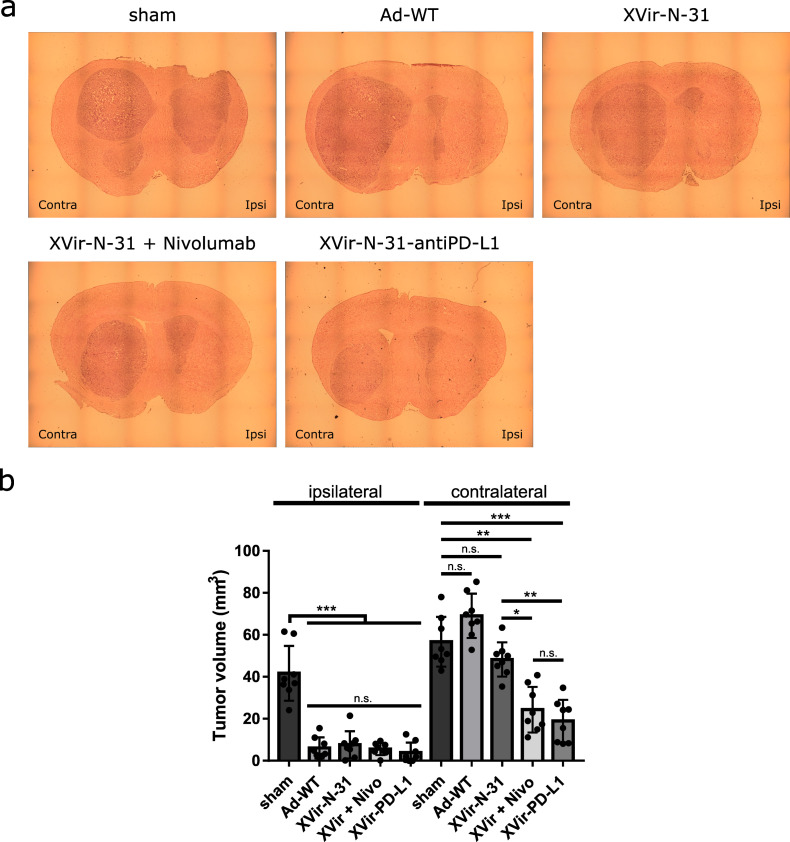
Reduction of tumor growth. (**a**) Representative pictures of brain sections of the U87MG tumor bearing, immuno-humanized NSG mice. The ipsilaterally growing tumor was injected with either PBS (sham), 3 × 10^8^ IFU Ad-WT, XVir-N-31, or XVir-N-31-anti-PD-L1. One group additionally received repeated systemic injections with nivolumab, as indicated in the Methods section. (**b**) Quantification of the tumor volume (mm^3^; XVir-PD-L1: XVir-N-31-anti-PD-L1; n = 7–8 mice per group, SEM; n.s.: not significant; * *p* < 0.05, ** *p*< 0.01, *** *p* < 0.001).

## Data Availability

The datasets generated during and/or analyzed during the current study are available on the Figshare data repository (https://doi.org/10.6084/m9.figshare.c.6174568, accessed on 28 August 2022).

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
