# Peer review of "The Oncolytic Adenovirus XVir-N-31, in Combination with the Blockade of the PD-1/PD-L1 Axis, Conveys Abscopal Effects in a Humanized Glioblastoma Mouse Model"

_ijms, 2022, doi:10.3390/ijms23179965_

Round 1

Reviewer 1 Report

Klawitter et al., have studied therapeutic efficacy of XVir-N-31 and XVir-N-31-anti-PD-L1in GBM models. For in vitro studies, OVs lytic activity was evaluated in L229 and U87MG cell lines. The results confirmed that (>30 MOI) lytic activity of OVs.

In this study, humanized glioblastoma mouse model was used to evaluate therapeutic efficacy. U87MG tumors on the ipsilateral as well as on the contralateral sides were created. Detection of the adenoviral hexon protein in ipsilateral virus-injected side was reported. However, on contralateral tumors adenoviral hexon protein not reported.

Furthermore, this study showed reduction of tumor growth and infiltration of immune cells in the ipsilateral and contralateral sides.

Comments

1.      Material and methods section: Please check numbering from 1.1.

2.      Conclusion can be placed before materials and methods section.

3.      There are eleven supplementary figures included; however, citation for Suppl Fig. 1 is not included in the main text. Please check.

4.      Legends for Supplementary figures are not included, please check.

5.      Mice were sacrificed at day 35. Was there any death occurred before day 35?

6.      Was there any a-PD-L1 related toxicity observed?

Author Response

Comments of reviewer 1:

  1. Material and methods section: Please check numbering from 1.1.

We checked the numbering of the subchapters in the material and methods part, however did not find mistakes. Since in the revised version of the manuscript the conclusions part has been placed before the material and methods section, the subchapters of the material and methods part are now numbered 5.1 to 5.12.

  1. Conclusion can be placed before materials and methods section.

We thank the reviewer to provides this idea since it generates a better overview on our study. We now placed the conclusion part before the material and methods section. In this regard we have also rewritten the conclusions part.

  1. There are eleven supplementary figures included; however, citation for Suppl Fig. 1 is not in  cluded in the main text. Please check.

We apologize that the citation was not visible on a first view. The citation for Suppl. Fig. 1. was included in the original manuscript in the material and methods part (original chapter 4.1. „Cells, cell lines and cell culture”; now chapter 5.1). As this supplementary figure describes the PD-L1 status of the used GBM cell lines, we believe it will be sufficient to place the citation at this position and not in the results part.

  1. Legends for Supplementary figures are not included, please check.

In the word and PFD file of the original manuscript that we uploaded, the legends for the supplementary figures were located at the far end of the manuscript (behind the reference list). We will take care that the legends of all figures and supplementary figures will also be included in the revised manuscript.

  1. Mice were sacrificed at day 35. Was there any death occurred before day 35?

No cell death in any mouse in any treatment group occurred before the time point we sacrificed all mice.

  1.  Was there any a-PD-L1 related toxicity observed?

We did not observe any anti-PD-L1 related toxicity in the mice that received an intratumoral injection of XVir-N31-anti-PD-L1, neither of those mice that received repeated injections of Nivolumab (anti-PD-1 antibody). All mice were checked and monitored regularly and in narrow time periods by us, the animal caretakers and by our veterinaries. The mice did not lose body weight nor showed any signs of toxicity. Additionally, administration of PD-1 or PD-L1 inhibitors in mice is already frequently published in dosages of 200µg/mouse or even higher (10mg/kg) without showing excessive toxicity or impaired survival. Please refer to:

  • Zeng J, et al. Int J Radiat Oncol Biol Phys. 2013;86(2):343-349.
  • Duraiswamy; , et al. Cancer Res(2013) 73 (23): 6900–6912.
  • Chandramohan et al. Journal for ImmunoTherapy of Cancer (2019) 7, Article number: 142 

Reviewer 2 Report

Oncoviral therapy is an emerging strategy for the treatment of tumors. In this manuscript, oncolytic adenovirus XVir-N-31 was investigated anticancer effect towards glioblastoma, in vitro and in vivo. In comparison, Ad-WT, XVir-N-31 expressing PD-L1 antibody, and XVir-N-31 combinatory with PD-L1 monoclonal antibody Nivolumab were included. In addition to lytic effect caused by virus infection, immunogenic cell death and immunostimulatory responses were determined. Overall, XVir-N-31 with PD-1/PD-L1 blockade produced remote effect on tumor regression.

The findings of this manuscript are of interests. However, the writing, data presentation, and designs should be improved.

1.     Abstract. “The oncolytic adenovirus (OAV) XVir-N-31 represents a promising therapeutic agent to treat glioblastoma (GBM)”. The rationale and concept should be described.

2.     Abstract. “Additionally, we combined XVir-N-31 with Nivolumab, or used XVir-N-31-anti-PD-L1”. The sentence is not clear.

3.     Please define OV.

4.     Please describe the model immunohumanized mouse model clearly and cited relevant references.

5.     “We have upgraded XVir-N-31 to express a PD-L1 blocking antibody and analysed the impact of XVir-N-31 and XVir-N-31-anti-PD-L1 on its immunostimulatory and therapeutic effects using immuno-humanized, orthotopic mouse GBM models.” Please describe clearly XVir-N-31-anti-PD-L1, producing antibody or PD-L1 binding protein.

6.     Figure 1. Cell lysis. Please indicate how to measure cell lysis.

7.     Figure 2. What type cells were used?

8.     Figure 2. Recombinant viruses were infected with 30 and 50 MOI. Why two different MOIs were used?

9.     YB-1 is a nuclear transcription factor. YB-1-dependent XVir-N-31 amplification is vital to XVir-N-31 application. Figure 3. The release of YB-1 was measured. Please describe the implication of YB-1 release.

10.  According to data of figure 3b, XVir-N-31-anti-PD-L1 caused surface presentation of CRT higher than other groups. However, the data were not explained and described in this manuscript.

11.  Figure 3. Mock control group should be included for comparison.

12.  The expression and surface presentation of CRT in tumor tissues should be determined because its importance in the actions of cytotoxic T cells.

13.  Data of figure 5 were prepared by coculture. Please describe why coculture system was required for this assay. Please discuss the sources for the expression of IFN-gamma.

14.  Please discuss how to translate this study to glioblastoma with low PD-L1.

Author Response

Reviewer 2:

Oncoviral therapy is an emerging strategy for the treatment of tumors. In this manuscript, oncolytic adenovirus XVir-N-31 was investigated anticancer effect towards glioblastoma, in vitro and in vivo. In comparison, Ad-WT, XVir-N-31 expressing PD-L1 antibody, and XVir-N-31 combinatory with PD-L1 monoclonal antibody Nivolumab were included. In addition to lytic effect caused by virus infection, immunogenic cell death and immunostimulatory responses were determined. Overall, XVir-N-31 with PD-1/PD-L1 blockade produced remote effect on tumor regression.

The findings of this manuscript are of interests. However, the writing, data presentation, and designs should be improved.

We thank the reviewer for his critical comments. We now prepared the figures using a higher resolution. Additionally, the manuscript was reviewed by a native English speaking person. 

  1. “The oncolytic adenovirus (OAV) XVir-N-31 represents a promising therapeutic agent to treat glioblastoma (GBM)”. The rationale and concept should be described.

We greatly thank reviewer 2 for this comment. We apologize that we did not explain the rationale and concept in the abstract, but we were limited by word counts. We have rewritten the abstracts part to improve the description, rationale, concept and outcome of our study. Therefore, we added some more information in the abstract part and hope this clarifies the above mentioned issues. We also hope that the editor will manage the oversized word count of the revised abstract.

  1. “Additionally, we combined XVir-N-31 with Nivolumab, or used XVir-N-31-anti-PD-L1”. The sentence is not clear.

We now added some more information in the abstracts part and hope this compensates the lack of clarity. We hope that the editor will manage the oversized word count of the revised abstract.

  1. Please define OV.

We apologize. We now added a few sentences and a reference in the introductions part to define oncolytic viruses. In addition, we added oncolytic virus (OV) the first time the abbreviation occurs in the text. Additionally, the abbreviation OV was explained in the abbreviation part.

  1. Please describe the model immunohumanized mouse model clearly and cited relevant references.

The immunohumanized mouse model was described shortly in the material and methods part of the original manuscript. To further explain this mouse model, we added some information in the material and methods part (now chapter 5.9) and added two references. However, by this addition the number of references that was limited to 50 by the publisher now reaches a number of >50. We hope that this can be managed by the editors.

  1. “We have upgraded XVir-N-31 to express a PD-L1 blocking antibody and analysed the impact of XVir-N-31 and XVir-N-31-anti-PD-L1 on its immunostimulatory and therapeutic effects using immuno-humanized, orthotopic mouse GBM models.” Please describe clearly XVir-N-31-anti-PD-L1, producing antibody or PD-L1 binding protein.

We described the cloning of the virus in the material and methods part of the original manuscript (chapter 4.2.). However, to give further and detailed information of how this virus was prepared we now added some more details in this chapter (new chapter 5.2.). However, the exact extensive cloning strategy can be provided on request.  Below a schematic depiction of the anti-PD-L1 construct is shown. We can add this into the manuscript on demand if the reviewer suggests us to do so.

  1. Figure 1. Cell lysis. Please indicate how to measure cell lysis.

We apologize that cell lysis induced by OVs might be unclear to be understood. Therefore, we now renamed the eradication of cells by virus replication “cell killing”. We now defined the term “oncolysis” in the text part. We determined oncolysis (the oncolytic mediated cell killing) on the one hand by taking photographs (Fig. 1A) that show the killing of cells and cell debris in OAV and Ad-WT infected cells. Dead or dying cells can be identified by their rounded shape. We added this sentence also in the legend of Fig. 1. Additionally, the killing of cells was determined after OAV infection using the MTT assay that measures the metabolic activity of cells and that is routinely used to determine cell viability (Fig. 1B). The method has been described in the material and methods part (former chapter 4.3, now chapter 5.3. in the revised manuscript).  We further added this information in the legend of Fig. 1 and changed the wording in different parts of the manuscript.

  1. Figure 2. What type cells were used?

We apologize to forget to give information about the type of cells used in this experiment. We used HEK293 cells as these cells are protein factories and allow to produce high amounts of recombinant proteins. We added this information in the legend of Fig. 2.

  1. Figure 2. Recombinant viruses were infected with 30 and 50 MOI. Why two different MOIs were used?

We thank the reviewer for the critical reading of our manuscript. Indeed, we used different amounts of virus for preparing supernatants and lysates for immunoblots (30 MOI, Fig. 2A) or for the functional assays (50 MOI, Fig. 2B). The rationale to use different MOI was the following:

For the functional test (Fig. 2B) we collected the supernatants earlier (48 h) than for immunoblots (48 h and 72 h). To be sure that on the one hand enough anti-PD-L1 was secreted by infected cells, in the functional experiment we decided to load the cells with a slightly higher amount of our OAVs (50 instead of 30 MOI). On the other side, we intended to avoid cell killing of infected cells by virus mediated lysis. Therefore, we collected the supernatants already 48 h after infection as we have observed (when preparing large scale virus) that an efficient oncolysis of HEK293 cells was observed around 60 – 80 hours after infection.

For the preparation of immunoblots we used only 30 MOI since we planned to demonstrate the expression and secretion of anti-PD-L1 at an early time point (48 h) when no or only very low cell lysis occurs as well as at a later time point (72 h, shortly before effective oncolysis of HEK293 cells occurs).

In summary: The use of a slightly higher infection rate (50 MOI) for functional testing was a compromise to (i) produce enough secreted anti-PD-L1, and (ii) to avoid high levels of virus-mediated cell killing (earlier time point of harvesting). We hope this explanation will answer the reviewer´s question to his satisfaction.

  1. YB-1 is a nuclear transcription factor. YB-1-dependent XVir-N-31 amplification is vital to XVir-N-31 application. Figure 3. The release of YB-1 was measured. Please describe the implication of YB-1 release.

It has been described that YB-1 will be secreted from cells during inflammatory processes. Besides, YB-1 was identified as a tumor-associated antigen capable to elicit a T cell response (reference 6). In this regard we were interested whether, besides the DAMPs HMGB1 and HSP70, also YB-1, that is expressed in GBM cells, will be released by OAV infected GBM cells. However, we have not analyzed the implication of YB-1 release in detail. Since YB-1 also is involved in virus replication, it would be a highly challenging task and will require substantial work to verify the immunological consequences of the release of YB-1. We are currently evaluating the possibility to initiate a research project addressing this issue. We added some sentences regarding this issue in the discussion part.

  1. According to data of figure 3b, XVir-N-31-anti-PD-L1 caused surface presentation of CRT higher than other groups. However, the data were not explained and described in this manuscript.

We thank the reviewer to pinpoint to this issue. We did also observe the higher CRT release by XVir-N-31-anti-PD-L1. Currently, we have no good explanation for this surprising finding. In this regard, it is also interesting that XVir-N-31-anti-PD-L1 shows a higher immunogenic cell death (although not significant) phenotype (Fig. 3) and IFN-γ release in U87MG cells (Fig. 5). It has been published that CRT is a dominant pro-phagocytic signal on multiple human cancer cells and is counterbalanced by CD47 (Chao et al Sci Transl. Med. 2010), and that a connection between CRT exposure and type I IFN signaling exists. This suggests that CRT functions as a “danger signal” that promotes a host type I IFN response associated with the induction of potent leukemia-specific T-cell immunity (Chen et al., Oncoimmunology 2017). This opens up interesting research options regarding the dynamic relationship between pro- and anti-phagocytic signals in human cancer and immune resistance cycle (Dosset et al Oncoimmunoloy 2018).

In this regard, initial results show an induction of CD47 expression in XVir-N-31 infected cells. If this correlates with CRT expression post virus infection is currently analyzed. However, our rather theoretical considerations so far need to be addressed by more relevant experiments.  Therefore, we added only a short comment on this issue in the results part (chapter 2.2) as well as in the discussion part.

  1. Figure 3. Mock control group should be included for comparison.

We apologize not to show the mock/sham treatment in Fig. 3A as we did for Fig. 3B, C and D.  We now show the mock treatment data also for Fig. 3A in the revised figure. As expected, the amount of HMGB1, HSP70 or YB-1 release from sham treated cells is as high as that from Ad-WT-infected cells

  1. The expression and surface presentation of CRT in tumor tissues should be determined because its importance in the actions of cytotoxic T cells.

We apologize not to show CRT surface expression in the tumors tissue, but we were not able to perform these stainings. Due to the large number of tumor tissue slices we needed for the quantification of several tumor infiltrating lymphocyte subtypes, for HMGB1, HSP70 and hexon stainings and for tumor volumetry, we were running out of material that can be used for these stainings.

  1. Data of figure 5 were prepared by coculture. Please describe why coculture system was required for this assay. Please discuss the sources for the expression of IFN-gamma.

We used cocultures as our goal was to determine the induction of IFN gamma that originates from both, tumor tissue infiltrating lymphocytes (in this experiment mimicked by the cocultured HLA-matched PBMCs) and from OAV infected GBM cells since this will reflect the in vivo situation in the tumor tissue. We added some additional explanation on this issue in the material and methods, the results as well as in the discussion part.

  1. Please discuss how to translate this study to glioblastoma with low PD-L1.

This is an interesting point. We added some sentences regarding this issue at the end of the discussion part.

Round 2

Reviewer 2 Report

The concerned comments had been answered.